# Characterization of the Secretome of Pathogenic *Candida glabrata* and Their Effectiveness against Systemic Candidiasis in BALB/c Mice for Vaccine Development

**DOI:** 10.3390/pharmaceutics14101989

**Published:** 2022-09-21

**Authors:** Majid Rasool Kamli, Jamal S. M. Sabir, Maqsood Ahmad Malik, Aijaz Ahmad

**Affiliations:** 1Department of Biological Sciences, Faculty of Science, King Abdulaziz University, Jeddah 21589, Saudi Arabia; 2Center of Excellence in Bionanoscience Research, King Abdulaziz University, Jeddah 21589, Saudi Arabia; 3Department of Chemistry, Faculty of Sciences, King Abdulaziz University, Jeddah 21589, Saudi Arabia; 4Department of Clinical Microbiology and Infectious Diseases, School of Pathology, Faculty of Health Sciences, University of the Witwatersrand, Johannesburg 2193, South Africa; 5Division of Infection Control, Charlotte Maxeke Johannesburg Academic Hospital, National Health Laboratory Service, Johannesburg 2193, South Africa

**Keywords:** secreted proteins, antifungal vaccine, *C. glabrata*, immunodominant antigens, LC–MS, BALB/c mice

## Abstract

Infections by non-albicans *Candida* species have increased drastically in the past few decades. *Candida glabrata* is one of the most common opportunistic fungal pathogens in immunocompromised individuals, owing to its capability to attach to various human cell types and medical devices and being intrinsically weakly susceptible to azoles. Immunotherapy, including the development of antifungal vaccines, has been recognized as an alternative approach for preventing and treating fungal infections. Secretory proteins play a crucial role in establishing host–pathogen interactions and are also responsible for eliciting an immune response in the host during candidiasis. Therefore, fungal secretomes can provide promising protein candidates for antifungal vaccine development. This study attempts to uncover the presence of immunodominant antigenic proteins in the *C. glabrata* secretome and delineate their role in various biological processes and their potency in the development of antifungal vaccines. LC–MS/MS results uncovered that *C. glabrata* secretome consisted of 583 proteins, among which 33 were identified as antigenic proteins. The protection ability of secretory proteins against hematogenously disseminated infection caused by *C. glabrata* was evaluated in BALB/c mice. After immunization and booster doses, all the animals were challenged with a lethal dose of *C. glabrata*. All the mice showing signs of distress were sacrificed post-infection, and target organs were collected, followed by histopathology and *C. glabrata* (CFU/mg) estimation. Our results showed a lower fungal burden in target organs and increased survival in immunized mice compared to the infection control group, thus revealing the immunogenic property of secreted proteins. Thus, identified secretome proteins of *C. glabrata* have the potential to act as antigenic proteins, which can serve as potential candidates for the development of antifungal vaccines. This study also emphasizes the importance of a mass-spectrometry approach to identifying the antigenic proteins in *C. glabrata* secretome.

## 1. Introduction

The frequency of invasive infections caused by pathogenic yeasts has increased globally, especially among immunocompromised patients. Due to the high mortality and morbidity rates associated with these pathogens, fungal infections are considered grievous [1]. *Candida* species account for bloodstream infections in immunosuppressed individuals in various hospital settings. The prevalence of *C. glabrata* has been observed in different geographical locations, making this pathogen second- to fourth-common fungal infection-causing agent [2,3]. Over the past few decades, candidemia has remained stable; however, a lower incidence of infections caused by *C. albicans* and a higher incidence of other non-albicans *Candida* species have been reported by several researchers. Among the various pathogenic non-albicans *Candida* species, *C. glabrata* is highly concerning, since it is intrinsically resistant to the azole class of antifungals [4,5]. Notably, there are reports of clinical strains of *C. glabrata* exhibiting low susceptibility to the echinocandin class of antimycotic drugs [6].

Phylogenetic analysis revealed that *C. glabrata* is related to *Saccharomyces cerevisiae* [7,8,9]. Therefore, the virulence attributes of *C. glabrata* seem to have progressed independently of other *Candida* species [9,10]. In accordance, neither hyphal formation nor secretory aspartyl proteases (Saps) has been reported from *C. glabrata* [8,10]. However, *C. glabrata* can potentially cause severe bloodstream infections as well as superficial mucosal infections in immunocompromised patients [11,12]. In addition, it has the potential to persist in both mouse and human macrophages [13,14,15].

Experts agree that introducing an anti-*Candida* vaccine will solve therapeutic complications caused by opportunistic fungal infections. It will also reduce the rate of occurrence and mortality caused by candidiasis. Therefore, developing an immune-based approach to combat such deadly infections is gaining attention. Remarkably, much effort has been dedicated to identifying immunodominant proteins from *C. albicans* and other non-albicans *Candida* species [16,17,18,19]. In addition, some researchers have also characterized the cell wall proteins and secretome of *C. glabrata* [20,21]. However, the role of these proteins as potential candidates for vaccine development against *C. glabrata* has not been explored yet. Many efforts have been invested into discovering the potential candidates for vaccine development against *C. albicans.* The major focus has been on exploring cell wall-associated proteins, as they play a critical role in host–fungal interaction, antigen presentation, and immunomodulation, which make them attractive vaccine candidates [22,23,24,25,26]. However, to date, no vaccines are available for infections caused by *Candida* species; thus, developing an anti-*Candida* vaccine is a significant priority. 

Regardless of therapeutic concerns, the pathogenic attributes of *C. glabrata* are unclear, and there are not enough reports about its immunodominant proteins. Therefore, analysis and characterization of the secretome of *C. glabrata* would be helpful for the identification of immunogenic proteins, which may serve as an essential tool for developing potential clinical biomarkers or candidates for vaccine development against candidiasis. Herein, we embarked on a study to uncover the protective effect of the secretome from pathogenic *C. glabrata* against systemic candidiasis in a murine model using BALB/c mice. Mass spectrometry was employed for the characterization of the secretome and to identify promising immunodominant proteins as prospective diagnostic biomarker/vaccine candidates. 

## 2. Materials and Methods

### 2.1. The Strains and Growth Conditions

This study used a control strain, *C. glabrata* ATCC2950, along with 3 clinical strains of *C. glabrata* obtained from various clinics of Johannesburg Academic Hospital, under the ethical clearance number M000402, Wits Human Research Ethics Committee. All the strains were revived and maintained for the experimental procedure on Sabouraud dextrose agar (SDA; Merck, RSA). 

### 2.2. Antifungal Susceptibility Profiling

Minimum inhibitory concentrations (MICs) of the antifungal drugs amphotericin B (AmB; Sigma-Aldrich, St. Louis, MO, USA) and caspofungin (CAS; Sigma-Aldrich, USA) against *C. glabrata* isolates were evaluated by the broth microdilution assay recommended in the standard M27 document (4th ed.) presented by the CLSI [27]. Briefly, stock solutions of the respective drugs were made in dimethyl sulfoxide (DMSO; Merck, Darmstadt, Germany, RSA), and test concentrations ranging from 16 to 0.008 µg/mL for AmB and CAS were prepared in 96-well flat-bottom microtiter plates followed by the addition of 100 μL yeast cells (10^6^ CFU/mL) and incubated at 37 °C for 24 h. Both sterile and growth controls were included in the experiment. 

The minimum fungicidal concentration (MFC) was calculated following MIC determination by spotting 20 µL from each well of 96-well plates on SDA and incubating them at 37 °C for 24 h. The lowest concentration with less than 5 colonies on the agar plate was recorded as the MFC [28]. Both MIC and MFC experiments were conducted two times in triplicate.

### 2.3. Screening of Pathogenicity Markers in Clinical Strains of C. glabrata

All four *C. glabrata* strains were investigated for crucial markers of pathogenicity. The existence of virulence factors, for instance, adherence, biofilm formation, secretory aspartyl proteinase (SAP), and phospholipase production, were checked for *C. glabrata* isolates. 

#### 2.3.1. Adherence Assay

Attachment of *C. glabrata* cells to oral epithelial cells (OPCs) was analyzed following the method described by Patel and co-workers [29]. Briefly, the OPCs were washed and re-suspended in 2 mL sterile PBS to a turbidity of 10^6^ cells/mL. Subsequently, *C. glabrata* cells (10^6^ cfu/mL) were grown at 37 °C for 24 h. After that, 2 mL each of yeast cells and OPCs were incubated for 2 h at 37 °C with shaking. Following incubation, the yeast and OPC mixture was filtered using 20 μm pore nylon filters (Sigma-Aldrich, Johannesburg, South Africa) to filter out non-adherent cells. The cell mixture was then washed and diluted in 1 mL dH_2_O followed by slide preparation and Gram staining. The yeast cells adhering to 100 OPC were counted under a light microscope for the adherence assay. 

#### 2.3.2. Biofilm Formation

Scanning electron microscopy (SEM) was utilized to confirm the ability of *C. glabrata* to form biofilms, and the method was followed as described previously [30]. Briefly, the yeast cells were grown on glass coverslips under standard biofilm growing conditions at 37 °C for 48 h. Post incubation, the culture broth was removed, and sessile cells were given a gentle wash with PBS followed by fixing with 2.5% glutaraldehyde for 2 h. Next, the fixed biofilms were carefully washed with PBS and then subjected to gradient dehydration with ethanol (40%, 10 min; 60%, 10 min; 80%, 10 min and 100%, 20 min). Later, the coverslips bearing fixed and dehydrated biofilms were subjected to critical point drying, carbon-coated, and observed under the SEM (Zeiss Gemini 2 Crossbeam 540 FEG SEM). 

#### 2.3.3. Hydrolytic Enzymes Production

*C. glabrata* strains were screened for the presence of hydrolytic enzymes by using methods described by Yousuf and co-workers [31]. Briefly, the test strain was grown for 18 h at 37 °C, followed by spinning the cells for 5 min at 3000 g and resuspending the cell pellet in fresh SDB. Post centrifugation, 2 μL aliquots were taken and spotted on proteinase agar plates and on egg yolk plates to determine SAP and phospholipase activity, respectively. After incubation at 37 °C for 3–4 days, the halo transparent zone around the colonies represented SAP activity, determined as Pz values, and was calculated by the ratio of the diameter of the colony to that of the diameter of the colony plus the clear zone of proteolysis. Similarly, the opaque zone surrounding the colonies on egg yolk plates represented the presence of phospholipase. The Pz values were evaluated in terms of the ratio of the diameter of the colony and the diameter of the colony plus the zone of precipitation.

### 2.4. Secretory Protein Extraction and LC–MS Analysis

#### 2.4.1. Extraction of Secreted Proteins from Growth Media

*C. glabrata* ATCC2950 was selected for further assays based on the antifungal susceptibility and pathogenicity markers. The secretory proteins from *C. glabrata* ATCC2950 were extracted by following the previously published protocols [16,32] with some modifications. Briefly, *C. glabrata* ATCC2950 cells were precultured overnight at 37 °C and 200 rpm in SDB growth medium and were later transferred to a yeast nitrogen base (YNB; Sigma-Aldrich, USA) supplemented with sucrose (20 g/L) at a concentration of 10^6^ cfu/mL and grown for 48 h. After incubation, cells were centrifuged, and the secured supernatant was filtered with the help of 0.4 μm membrane filters to eliminate any remaining cells. Secreted proteins were then precipitated by adding chilled acetone supplemented with 10% TCA for overnight at 4 °C. Following spinning (12,000 rpm, 4 °C) for 1 h, the supernatant was thrown away, and the precipitated proteins were washed three to four times with chilled acetone. The remaining protein pellet was secured, air-dried for 30 min, and resuspended in lysis buffer. The final concentration of protein present in the solution was estimated using the DC protein assay (Bio-Rad, Hercules, CA, USA), with BSA as a standard, following the manufacturer’s guidelines.

#### 2.4.2. Sample Preparation for LC–MS Analysis

Samples of the secretory fraction of *C. glabrata* ATCC 2950 were incubated with Bolt™ LDS sample buffer and reducing agent (Invitrogen, Waltham, MA, USA) and electrophoresed on a Bolt™ 4 to 12%, Bis–Tris, 1.0 mm Mini Protein Gel (Invitrogen, USA). The polyacrylamide gels were stained with GelCode™ Blue Stain Reagent (Thermo Fisher Scientific, Waltham, MA, USA) and destained with Milli-Q water. Based on Shevchenko and co-workers, proteins were digested from gel fractions [33]. Each sample was prepared in 6 separate gel fractions (based on approximated molecular weight ranges: >100 kDa, 60–100 kDa, 40–60 kDa, 25–40 kDa, 15–25 kDa, and <15 kDa). Briefly, the proteins were reduced in gel with 10 mM DTT in 25 mM AMBIC for 1 h at sixty degrees. Samples were cooled to room temperature, and then 100% acetonitrile was added and incubated for 10 min followed by discarding of the supernatant and addition of 55 mM iodoacetamide (IAA) in 25 mM AMBIC to the gel pieces. The reaction progressed for 20 min at room temperature in a dark room. The supernatant was discarded, gels were dehydrated with 25 mM AMBIC in 50% acetonitrile and vortexed, and the supernatant was discarded followed by drying to completeness, and freshly prepared trypsin was added. Protein digestion was allowed to proceed overnight at 37 °C. The digestion was quenched by adding formic acid to a final of 0.1%, and the samples were dried under a vacuum. For mass spectrometry analysis, the dried samples were re-suspended in 2% acetonitrile and 0.2% formic acid. The mass spectrometry analysis was conducted three times to ensure the reproducibility of the data. 

#### 2.4.3. LC–MS Data Acquisition

Tryptic peptides from each gel fraction were analyzed using a Dionex 3000 RSLC system combined with an AB Sciex TOF mass spectrometer. With the help of an Acclaim PepMap C18 trap column (75 μm × 2 cm; 2 min at 5 μL min^−1^ using 2% ACN/0.2% FA), all the injected peptides were de-salted. All the trapped peptides were then eluted by gradients and separated using a Waters nanoEase CSH C18 column (75 μm × 25 cm, 1.7 µm particle size) with a 0.3 µL min^−1^ flow-rate and a gradient of 10–55% B over 10 min (A: 0.1% FA; B: 80% ACN/0.1% FA). The 6600 Triple TOF mass spectrometer was operated in positive ion mode. Data-dependent acquisition (DDA) was employed; precursor (MS) scans were acquired from m/z 400–1500 (2 ± 5 + charge states) using an accumulation time of 100 ms followed by 40 fragment ion (MS/MS) scans, acquired from m/z 100–1800 with 20 ms accumulation time each.

#### 2.4.4. LC–MS Database Searching

Raw data files (.wiff) were searched with Protein Pilot V5.0 software (SCIEX), using a database containing sequences from the *C. glabrata* reference proteome (downloaded from UniProt on 15 November 2021) and the usual impurities. Trypsin was set as the digestion enzyme, cysteine alkylation (iodoacetamide) was allowed as a fixed modification, and biological modifications were allowed in the search parameters. A 1% false-discovery rate filter was applied at the protein level for refinement of identifications.

#### 2.4.5. Bioinformatic and Functional Analyses

The protein sequences (ORF translation) were obtained from the *Candida* Genome Database (http://www.candidagenome.org/ (accessed on 15 November 2021) by the gene names for the identified proteins [34] and using the CGD Batch Download tool and gene ontology enrichment analysis using the CGD GO Slim Mapper tool with the default settings. The complete list of sequences was submitted to the VaxiJen v2.0 server (http://www.ddg-pharmfac.net/vaxijen/VaxiJen/VaxiJen.html (accessed on 15 November 2021)) [35]. To predict the presence of protective antigens, a threshold score of 0.9 was applied. The SignalP-5.0 (https://services.healthtech.dtu.dk/service.php?SignalP-5.0 (accessed on 15 November 2021) [36] and DeepLoc-1.0 (https://services.healthtech.dtu.dk/service.php?DeepLoc-1.0 (accessed on 15 November 2021) [37] servers were used to identify the presence of secretory signal peptide and accurately (profile setting) predict subcellular localization of the predicted antigens, respectively. 

### 2.5. In Vivo Studies

In the present study, female BALB/c mice (8–10 weeks) with an approximate weight of 18–21 g were engaged. All the experiments were performed per the guidelines by the Wits Animal Research Ethics Committee (AREC-2019/04/20/c). The mice were allowed to acclimatize for one week before starting the experiments (Figure 1).

### 2.6. Immunization Protocols

The animals were divided into three groups: (*a*) immunized with secretome group (n = 10), (*b*) unimmunized and uninfected (healthy control) group (n = 10), and (*c*) unimmunized and infected (infection control) group (n = 10). A dose to 60 μg of secretome was used to immunize each mouse in a designated group through the intraperitoneal route without any adjuvant system. Before immunization (day 0) and post-immunization, on days 7, 14, and 21, blood samples were collected through the tail vein from each group of mice. On day 21, a booster dose was administered with the same protein concentration and route. Two weeks after the booster dose, blood was again collected from all groups, followed by challenging each mouse through the tail vein with a dosage of 5 × 10^7^ CFU/mouse (as estimated by using MicroScan Turbidity meter, Beckman Coulter, in a total volume of 150 μL of sterile saline) of *C. glabrata* ATCC2950. A plate count confirmed the cell count and viability of infecting inoculum. The protective effect of immunization was examined by monitoring the survival rate of vaccinated mice after infection (up to three weeks). 

### 2.7. Histopathologic Examination

After monitoring the survival profile in both vaccinated and unvaccinated groups, all the animals were sacrificed according to the SOP followed by the animal house. After euthanization, kidneys, liver, lungs, and spleen were removed aseptically, weighed, and fixed in 10% buffered formalin. All the fixed tissues were implanted in paraffin, followed by cutting longitudinal sections of 4 µm and staining them with periodic acid–Schiff (PAS). The stained sections were observed under a microscope (Leica Microsystems, Heerbrugg, Switzerland). 

For determination of the fungal burden, all tissues were homogenized in PBS (chilled), and the resulting homogenate was seeded onto SDA plates with 100 μg/mL ampicillin, followed by incubation at 37 °C for 48 h, and later on, colonies were counted, and the result recorded as log_10_ CFU/mg of tissues. 

### 2.8. Statistical Analysis

The survival rate and fungal burden results were analyzed using a log-rank test and two-tailed Student’s *t*-test in GraphPad Prism 9.3.1 (350) software (GraphPad Software, San Diago, CA, USA). *p* values < 0.05 were considered statistically significant.

## 3. Results and Discussion

### 3.1. Antifungal Susceptibility Profiling

The antifungal susceptibility of *C. glabrata* strains against the commonly used antifungals, Am B and CAS, was obtained from the broth microdilution assay, and MIC and MFC values for Am B and CAS are recorded in Table 1. Based on the MIC values, all the strains appeared to be susceptible to all the three tested antifungal drugs, with *C. glabrata* ATCC2950 having the highest MIC values.

### 3.2. Pathogenicity Markers 

#### 3.2.1. Adherence 

Adherence to host cells is the initial pathogenicity marker of *Candida* species and was recorded as the number of yeast cells attached to each of the 100 OPCs. The results obtained demonstrated that all the four strains of *C. glabrata* showed the ability to adhere to OPCs to varying degrees. Compared to clinical isolates, *C. glabrata* ATCC2950 adhered to OPCs at a higher rate with a mean value of 271 ± 11; whereas for other *C. glabrata* isolates, the average number of yeast cells was 187 ± 33. 

#### 3.2.2. Biofilm Formation

All the strains have the capability of forming biofilm, but the extent varied among the strains. Based on the results, *C. glabrata* ATCC2950 were found to have higher metabolic activity and therefore higher biofilm-forming capability than the other three *C. glabrata* strains (Figure 2). 

Additionally, this strain had higher MIC values than the other tested strains, which could be linked to biofilms.

#### 3.2.3. Hydrolytic Enzymes

All the *C. glabrata* isolates were tested for the secretion of aspartyl proteinase and phospholipase enzymes, which are the characteristic pathogenicity markers of candidiasis. All the isolates were positive for proteinase production, with Pz values ranging from 0.28 to 0.42 (Figure 3A). However, all the tested *C. glabrata* strains tested negative for phospholipase activity with Pz value = 1 (Figure 3B).

Based on these results, *C. glabrata* ATCC2950 was found to be the most virulent strain with the highest MIC values against all the tested antifungals, as well as displaying increased pathogenic attributes, and therefore we selected *C. glabrata* ATCC2950 for further experimental analysis.

### 3.3. Proteomic Analysis of C. glabrata Secretome 

In this study, the global protein composition of the *C. glabrata* secretome was analyzed, and 583 groups of protein were identified by LC–MS. These proteins were subjected to gene ontology enrichment analysis (CGD GO Slim Mapper tool). The results showed a high frequency of proteins involved in transport (110 proteins, 18.9%), translation (100 proteins, 17.2%), regulation of biological processes (100 proteins, 17.2%), organelle organization (97 proteins, 16.6%), response to stress (67 proteins, 11.5%), and carbohydrate metabolic process (64 proteins, 11%). The enrichment distribution of the identified proteins of the *C. glabrata* secretome is presented in Figure 4.

*Candida* species cause deep-seated and systemic infection in humans by circumventing the host defense system and adapting to the changing host microenvironment. Secreted proteins play a significant role in the pathogen’s ability to cope with the challenges faced inside the host, their pathogenicity, acquiring nutrition, and evasion of the immune defense. Numerous studies have shown the importance of extracellular hydrolytic enzymes [38,39]; however, some members of the families of phospholipases, lipases, and aspartyl proteases are usually not reported under laboratory conditions, whereas they have been found in abundance in vivo [40]. This finding is because in vitro conditions cannot truly mimic the host environment during infection. The pathway of secreted proteins is highly regulated and only observed in the host. Although several researchers have also supported this unusual behavior of secretory proteins, only around 12% of these proteins have been spotted under all laboratory conditions, whereas more than 30% of the secreted proteins were identified only under one particular in vitro condition [16,40,41]. Secreted proteins have a critical role in establishing infection inside the host. Simultaneously, some also produce a robust immune response; however, at one given time, only a subset of these essential but highly regulated proteins are secreted [41]. 

Rasheed et al. reported a functional enrichment analysis of the secretome in wild type *C. glabrata* and found proteins predominant in cell wall organization, proteolysis, and translation; whereas the secretome from *C. glabrata* mutant (*Cgyps1-11*Δ, also referred to as CgYapsins) was found enriched in transport, de novo co-translational protein folding, pentose-phosphate shunt, and removal of superoxide radicals’ biological processes. Although similar protein groups have been reported previously by other workers, our data include the presence of a few crucial groups of proteins such as response to stress and chemicals in the secretome, therefore advocating the importance of secreted proteins in adaptation to unfavorable host microenvironments and causing infection in the host. 

### 3.4. Functional Analysis of Predicted Antigens of C. glabrata Secretome

Considering the critical role of secreted proteins in host–pathogen interactions during the course of infection, identifying these proteins and uncovering their immunogenic potential will be beneficial for designing new therapeutic biomarkers and developing vaccine strategies against *Candida* infection [42,43]. All the identified proteins were submitted to the VaxiJen v2.0 server to predict the presence of protective antigens. In contrast, the secretory signal peptide and their subcellular localization were validated using SignalP-5.0 and DeepLoc-1.0 servers, respectively. Of the total of 583 proteins identified, 33 were predicted to be protective antigens. The predicted antigens were annotated by determining the presence of secretory signaling peptide sequences and the subcellular protein localization (Table 2). Out of 33 antigenic proteins identified, Bgl2, Cwp1, and Awp1 proteins were flagged with secretory signaling peptides and confined to the extracellular region, whereas Fpr2 was found restricted in the endoplasmic reticulum.

*Candida* species secrete various proteins that play a key role in the mortification of host proteins/carbohydrates/lipids, and the acquirement of essential ions (zinc and others) [43,44]. The proteins can either be secreted by the classical secretory pathway or by non-conventional pathways [45]. Researchers have demonstrated the secretion of some typical cytoplasmic proteins (devoid of a signal peptide) inside extracellular vesicles (EV) [42,43], and a few of them have been named as “moonlighting” proteins, since they hold diverse properties based on their subcellular positions [46,47]. Twenty-one proteins (bold) out of 33 predicted antigenic proteins have been previously characterized in the *C. glabrata* secretome [21], but their immunodominant nature was not evaluated, whereas the rest were reported for the first time as a part of the *C. glabrata* secretome in the present study. Extensive research has been done to identify new biomarker candidates for invasive candidiasis using cell walls or intracellular proteins from *C. albicans* [48,49,50]. Proteins such as Bgl2 and Asc1, also recognized in the present investigation, were also described in the hyphal secretome of *C. albicans*, which were acknowledged to be antigenic [51]. Similarly, the functional relevance of Asc1 proteins (a component of the 40S ribosomal subunit) were reported earlier, and their role in cellular adhesion and pathogenicity by regulating the expression of specific genes in *C. albicans* was discussed in detail [52].

Dolichol phosphate mannose synthase (DPMS) plays a vital role in cell wall composition and morphogenesis in *C. albicans* [53]. Similarly, Pnc1 protein, a product of the *PNC1* gene, has been reported to play a critical role in cellular aging, as yeast cells consisting of higher copies of this gene have a longer life span (70%) than the wild type strain [54], whereas Scp160 protein plays an essential role in yeast translation [55], and adhesin-like proteins, Awp1, are involved in adherence to human epithelial and biofilm formation [20]. Furthermore, the role of other predicted antigenic proteins, such as SGTA dimer domain-containing protein and so on, needs further in-depth validation for their role in the *C. glabrata* secretome. Since much is not reported about the antigenic nature of secreted proteins in *C. glabrata*, still, variability in the composition of the secretome may result in a significant difference in pathogenicity between various species of *Candida* as well as variations in the host–pathogen interactions, severity of infection caused, and more significant problems in the treatment of such fungal infections. 

Hence, there is an urgent need to uncover and analyze the secretome of *C. glabrata*. Therefore, the present study gives a better insight into the antigenic proteins present in the *C. glabrata* secretome that may be targeted for inventing new treatments against this pathogenic species of *Candida.* Numerous candidate antigens in secretory proteins that had not previously been reported in this form were identified in this study. Some of the present proteins have unknown functions, while others play roles in aerobic respiration and are ribosomal proteins, such as Rpl10, Rpl12B, Rpl26A, Rpl32, Rpl33A, Rps8A, Rps21A, Rps24A, and Rps29B. Additionally, Rpl and Rps classes of proteins have been described to be linked to the cell surface of *C. albicans* and involved in the stress response [56]. Therefore, these novel antigenic proteins may act as an unambiguous biomarker for *C. glabrata*. In addition, the existence of proteins such as Bgl2, Asc1, and DPMS, reported to have immunomodulatory properties in *C. albicans*, suggest that these proteins can target both *C. glabrata* and *C. albicans.* However, further studies are required to advocate these claims.

### 3.5. In Vivo Studies

#### Effectiveness of Secreted Proteins against *C. glabrata* Infection 

*C. glabrata* is predominantly responsible for invasive infections associated with significant mortality in immunocompromised individuals. As drug resistance to this pathogen has significantly increased in recent years, it is paramount to identify alternative approaches to tackle such pathogens. Identifying different vaccine candidates has gained importance in preventing different infections; unfortunately, there is no approved vaccine to prevent *C. glabrata* infections. In the present study, the secretome from *C. glabrata* ATCC2950 was used to immunize BALB/c mice to prevent the infection caused by *C. glabrata*. Infection with a dose of 5 × 10^6^ CFU/mouse of *C. glabrata* ATCC2950 caused high morbidity in BALB/c mice with very high fungal burdens in liver, spleen, kidney, and even in lungs. 

To test the protection capability of secreted proteins, mice were vaccinated with the secretome in two doses (primary + booster), and then challenged with *C. glabrata* through the tail vein. Despite higher doses of infection with pathogenic *C. glabrata*, 100% mortality was never achieved in control groups; however, mice in the infection control group displayed signs of distress, such as orbital tightening, rough hair coat, and gradual weight loss, after 24 h of infection. These results agree with the previous findings by Hirayama and co-workers (2020) that *C. glabrata* infection in immunocompetent mice does not lead to 100% mortality [57].

In comparison to the infection control group, the immunized mice showed minor signs of distress, such as uneven hair coat. After 24 h of infection, four mice from the infection control group and two mice from the immunized group were showing signs of distress and were therefore euthanized, whereas after 48 h of infection, two mice from both these groups were sick and were euthanized (Figure 5). 

All the remaining animals were observed until the end of the experiment (21 days). Immediately after sacrificing, the target organs were removed, weighed, and processed for fungal burden calculation and histopathology. The degree of infection was estimated in terms of fungal burden. The infection control group observed a high fungal burden in the liver, spleen, kidney, and lungs. In contrast, the fungal burden in all studied organs of immunized animals was found to be significantly low (Figure 6). 

Histopathology was carried out, and the results confirmed the protective ability of secreted proteins against fungal infection, evident from the lower fungal burden in all the tested organs of vaccinated BALB/c mice. In contrast, the mice from the infection control group showed several micro-abscesses, with *C. glabrata* cells distributed throughout the organ (Figure 7). It is important to note that in Figure 7, kidneys are showing higher fungal burden, which is not consistent with the results shown in Figure 6, where higher CFU/mg was seen in spleen. This inconsistency can be related to the fact that Figure 7 is a representative image and only focuses on a small section of slide, whereas CFU count is the total count of cells present in the organ. 

Low CFU/mg and histopathology confirmed the preventive efficacy of the secretome against *C. glabrata* infection. The average log reduction in CFU/mg in immunized mice was around 1.4 for all target organs compared to infection control mice. The confirmed presence of 33 different immunogenic proteins is predicted to help the hosts to create antibodies that help them fight disseminated infections. In addition, the virulence-associated proteins in the *C. glabrata* secretome further strengthen its immunogenic property. 

In addition to CFU/mL and histopathology, all the organs were analyzed for any characteristic changes. It was noted that there was a difference in the weight of the target organs in different groups. The healthy control’s average weights of liver, spleen, kidneys, and lungs were 1.16 ± 0.01 g, 0.13 ± 0.02 g, 0.15 ± 0.03 g, and 0.16 ± 0.01 g, respectively. In comparison, the organs in the infection control group were found inflamed and heavy on the scale, whereas the immunized mice had low inflammation compared to the infection control (Figure 8). These results further substantiate the in vivo potential of the antigenic proteins present in the *C. glabrata* secretome in protecting against *C. glabrata* infection by possibly eliciting a humoral immune response. 

In a similar study conducted by Hirayama and co-workers (2020), *C. glabrata* was only found in the liver and kidneys [57], whereas the results in this study revealed high *C. glabrata* burden in the liver, kidney, and spleen. Furthermore, researchers have reported the capability of *C. glabrata* to replicate and survive in macrophages [58], and monocytes may facilitate the dissemination of *C. glabrata* and protect it from host extracellular defenses [59]. However, there are no reports on the accumulation of *C. glabrata* cells in the spleen and lungs. The ability of *C. glabrata* to survive in various organs for a prolonged time period, despite the presence of an active host immune system, was also supported previously by other researchers [13,60,61,62].

This work identified numerous candidate antigens in secretory proteins, which had not previously been reported in this form. Some of the present proteins have unknown functions, while others play a role in aerobic respiration and are ribosomal proteins, such as Rpl10, Rpl12B, Rpl26A, Rpl32, Rpl33A, Rps8A, Rps21A, Rps24A, and Rps29B. Additionally, the Rpl and Rps class of proteins has been found to be linked with the cell surface of *C. albicans* and involved in the stress response [56]. These novel antigenic proteins may act as an unambiguous biomarker for *C. glabrata*. In addition, exitance of proteins such as Bgl2, Asc1, and DPMS, reported to have immunomodulatory properties in *C. albicans,* suggests that these proteins can be used to target both *C. glabrata* and *C. albicans*; however, further in-depth study will determine the candidature of these predicted antigenic proteins for therapeutics.

## 4. Conclusions

This pilot study successfully identified the presence of immunogenic proteins in the secretome of *C. glabrata* ATCC2950 using mass spectrometry. The results provide insight into the immunogenic secretory proteins and immune pathogenesis of *C. glabrata.* These antigenic proteins may serve as potential targets for novel therapeutic strategies, including candidates for diagnostic and/or vaccine development against *Candida* species. However, further work will expand upon this study to identify pure antigenic proteins and their applications in antifungal vaccine development. 

## Figures and Tables

**Figure 1 pharmaceutics-14-01989-f001:**
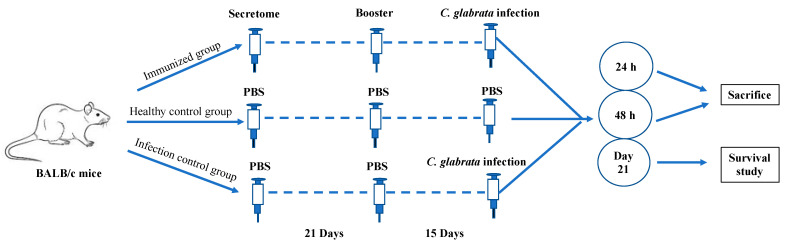
*Candida glabrata* secretome-immunized mice are protected from *C. glabrata* infection. BALB/c mice were vaccinated twice with either secretome or PBS alone. After 15 days of the booster, the mice were infected with a lethal dose of *C. glabrata* through the tail vein. Each group consisted of 10 mice; post-infection, animals showing signs of distress were sacrificed after 24 h and 48 h, whereas the rest were observed for the survival study until 21 days.

**Figure 2 pharmaceutics-14-01989-f002:**
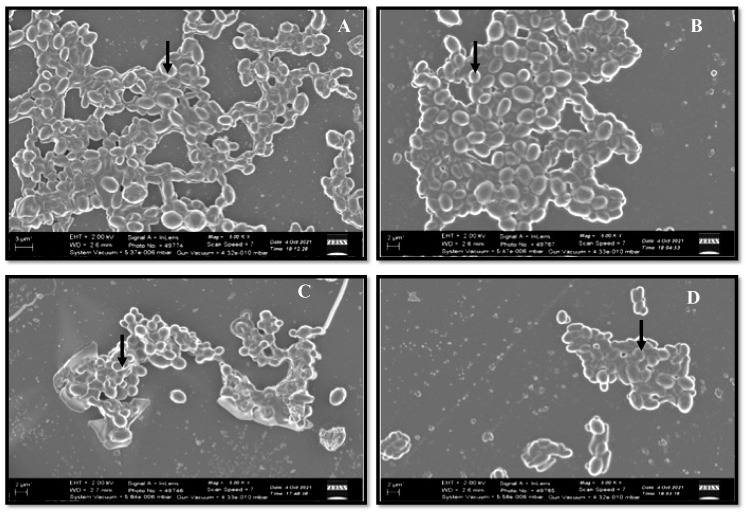
Representative SEM images of biofilms of different *C. glabrata* isolates. The cells were grown under biofilm-forming conditions, and the biofilm architecture was analyzed using SEM. (**A**) *C. glabrata* ATCC2950, (**B**) *C. glabrata* 3471, (**C**) *C. glabrata* 3469, (**D**) *C. glabrata* 3471. The arrows show *C. glabrata* cells embedded in the biofilm matrix.

**Figure 3 pharmaceutics-14-01989-f003:**
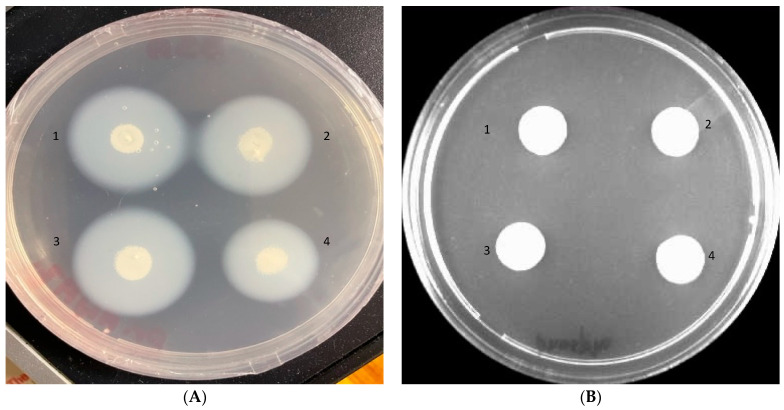
Proteinase (**A**) and phospholipase (**B**) activities of different *C. glabrata* isolates on BSA and egg yolk plates, respectively. A zone of clearance around the colonies (**A**) indicates positive proteinase activity, while no precipitation zones around the colonies (**B**) indicates negative phospholipase activity. (1) *C. glabrata* ATCC2950; (2) *C. glabrata* 3411; (3) *C. glabrata* 3469; and (4) *C. glabrata* 3471.

**Figure 4 pharmaceutics-14-01989-f004:**
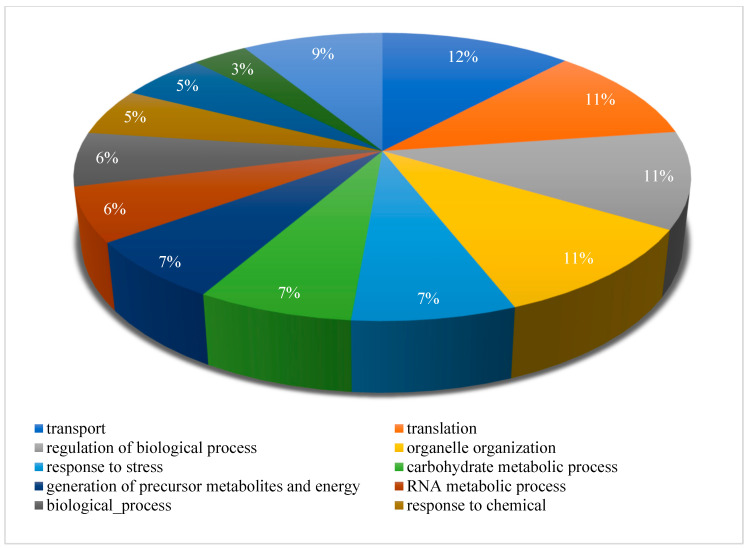
Pie distribution chart showing gene ontology enrichment analysis of biological processes for the proteins identified in the *C. glabrata* secretome. Biological processes that were common to <5% of the proteins were excluded from the chart.

**Figure 5 pharmaceutics-14-01989-f005:**
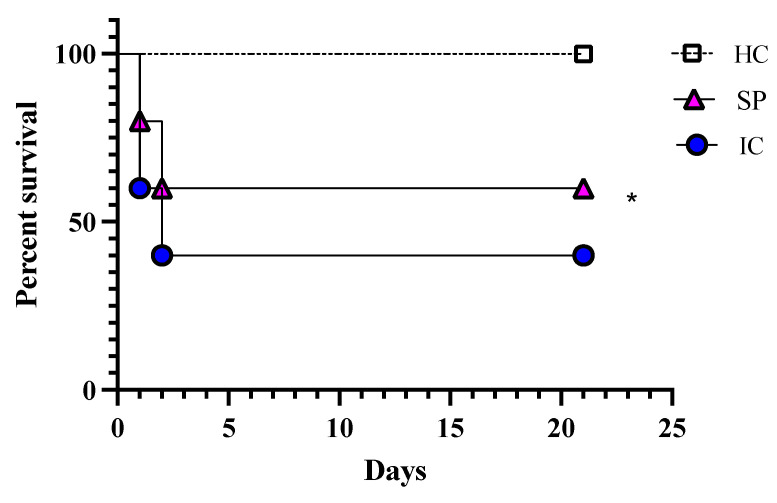
Survival probability of *C. glabrata* secretome-immunized mice. The survival proportion states the potency of secreted proteins in protection against infection caused by *C. glabrata* compared to the un-immunized group. A Mantel–Cox log-rank test (n = 10 mice/group) was used to compare the survival of mice in different groups. HC, healthy control group; SP, secreted protein; IC, infection control. * *p* = 0.0171.

**Figure 6 pharmaceutics-14-01989-f006:**
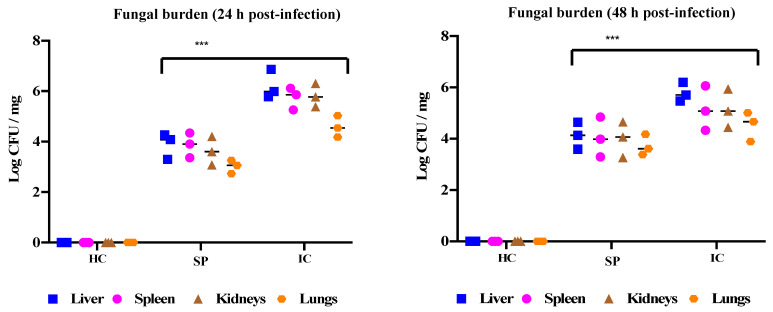
The protective effect of the *C. glabrata* secretome against fungal burden. Post infection immunized and infection control mice were sacrificed, and liver, spleen, kidneys, and lungs were harvested and subjected to fungal burden determination. The fungal burden was expressed as geometric means of logarithmic values for CFU/mg of tissues. HC, healthy control group; SP, secreted proteins; IC, infection control. *** *p* = 0.0043.

**Figure 7 pharmaceutics-14-01989-f007:**
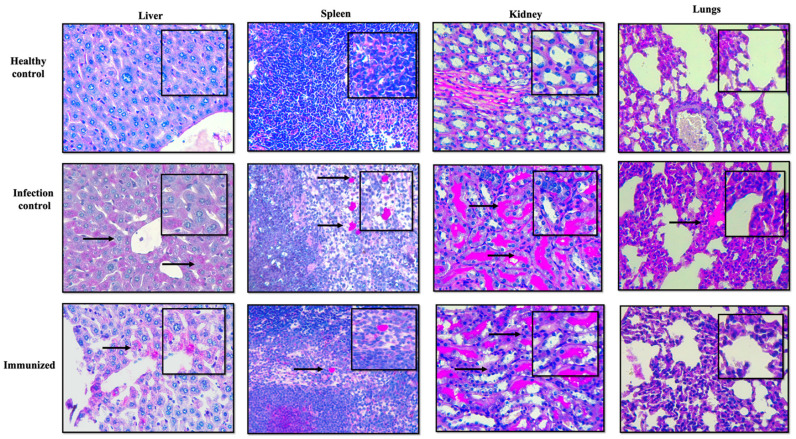
Comparative histopathology analysis of different tissue sections from immunized and *C. glabrata*-infected mice. Tissue sections recovered post infection were stained with PAS. The mouse from the infection control group displayed numerous abscesses with visible *C. glabrata* cells throughout the tissues, whereas secreted protein vaccinated mouse did not show any visible abscesses with a highly reduced fungal burden. The arrows indicate the presence of *C. glabrata* cells in the target tissues.

**Figure 8 pharmaceutics-14-01989-f008:**
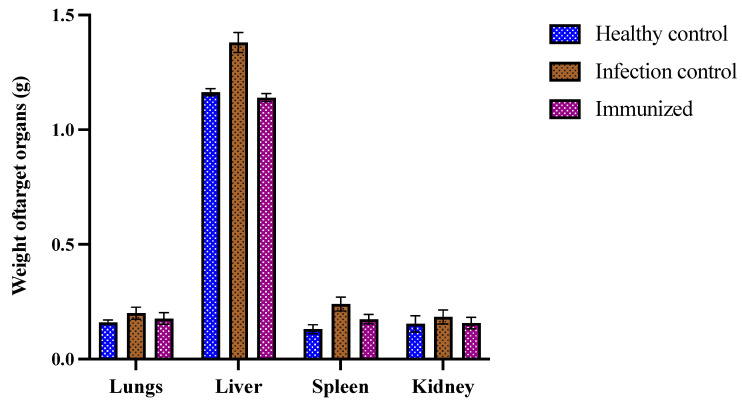
Average weight of target organs (g). Post-sacrifice, the target organs were removed and weighed to study the difference in weight between organs (lungs, liver, spleen, and kidney) among healthy, infected, and immunized groups.

**Table 1 pharmaceutics-14-01989-t001:** Antifungal susceptibility profile of *C. glabrata* strains.

Species	MIC/MFC (µg/mL)			
	Am B		CAS	
	MIC	MFC	MIC	MFC
*C. glabrata* ATCC2950	1	2.0	0.5	2.0
*C. glabrata* 3411	0.5	1.0	0.25	1.0
*C. glabrata* 3469	0.125	0.5	0.25	1.0
*C. glabrata* 3471	0.125	0.25	0.25	1.0

Amphoterecin B, AmB; caspofungin, CSA; minimum inhibitory concentration, MIC; minimum fungicidal concentration, MFC.

**Table 2 pharmaceutics-14-01989-t002:** List of predicted antigenic proteins present in the *C. glabrata* secretome.

Sr. No.	*C. glabrata* ID	Common Name	Subcellular Localization	Soluble/Membrane	Secretory Peptide
1.	CAGL0A01716g	Pnc1	Cyto	Sol	No
2.	**CAGL0A04521g**	**Rps8A**	**Cyto**	**Sol**	**No**
3.	**CAGL0D00858g**	**Rps29B**	**Cyto**	**Sol**	**No**
4.	**CAGL0D02090g**	**Asc1**	**Nuc**	**Sol**	**No**
5.	CAGL0E02739g	SGTA_dimer domain-containing protein	Cyto	Sol	No
6.	**CAGL0F02937g**	**Rpl12B**	**Cyto**	**Sol**	**No**
7.	CAGL0F04477g	Proteasome subunit beta type-6	Cyto	Sol	No
8.	**CAGL0G00220g**	**Bgl2**	**ExC**	**Sol**	**Yes**
9.	**CAGL0G01078g**	**Rpl26A**	**Cyto**	**Sol**	**No**
10.	CAGL0G05357g	NAD(P)H-hydrate epimerase	Cyto	Sol	No
11.	CAGL0G09955g	Dolichol-phosphate mannosyltransferase subunit 1 (DPMS)	ER	Memb	No
12.	**CAGL0H01705g**	**Fpr2**	**ER**	**Sol**	**Yes**
13.	**CAGL0H02057g**	**Gar1**	**Nuc**	**Sol**	**No**
14.	CAGL0H02321g	Coupling of ubiquitin conjugation to ER degradation protein 1	ER	Memb	No
15.	**CAGL0H04521g**	**Rpl32**	**Cyto**	**Sol**	**No**
16.	CAGL0H07023g	ATP synthase subunit e, mitochondrial	Mito	Memb	No
17.	**CAGL0H08541g**	**Nhp6B**	**Nuc**	**Sol**	**No**
18.	**CAGL0I00594g**	**Gsp1**	**Cyto**	**Sol**	**No**
19.	CAGL0I07969g	ATP synthase subunit K, mitochondrial	Mito	Memb	No
20.	**CAGL0J01463g**	**Cwp1**	**ExC**	**Sol**	**Yes**
21.	CAGL0J02508g	Awp1	ExC	Memb	Yes
22.	**CAGL0J03234g**	**Rps24A**	**Cyto**	**Sol**	**No**
23.	**CAGL0J07678g**	**Sui1**	**Cyto**	**Sol**	**No**
24.	**CAGL0K06281g**	**Guk1**	**Cyto**	**Sol**	**No**
25.	CAGL0K07744g	Nudix hydrolase domain-containing protein	Mito	Sol	No
26.	**CAGL0K08382g**	**Rps21A**	**Cyto**	**Sol**	**No**
27.	**CAGL0K11572g**	**Sba1**	**Nuc**	**Sol**	**No**
28.	**CAGL0K12826g**	**Rpl10**	**Cyto**	**Sol**	**No**
29.	**CAGL0M02497g**	**Rpl33A**	**Cyto**	**Sol**	**No**
30.	CAGL0M03223g	Protein SCP160	Cyto	Sol	No
31.	**CAGL0M04983g**	**Mbf1**	**Nuc**	**Sol**	**No**
32.	CAGL0M12595g	Actin-related protein 2/3 complex subunit 5	Cyto	Sol	No
33.	**CAGL0M13277g**	**Ynk1**	**Cyto**	**Sol**	**No**

Cytoplasm = Cyto; Nucleus = Nuc; Mitochondrion = Mito; Extracellular = ExC; Endoplasmic reticulum = ER; Soluble = Sol; Membrane = Memb. The predicted antigenic proteins represented in bold font have been previously characterized in the *C. glabrata* secretome, whereas the rest were reported for the first time as a part of the *C. glabrata* secretome in the present study.

## Data Availability

The mass spectrometry proteomics data have been deposited to the ProteomeXchange Consortium via the PRIDE [1] partner repository with the dataset identifier PXD032928.

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
