# Peer review of "Characterization of the Secretome of Pathogenic *Candida glabrata* and Their Effectiveness against Systemic Candidiasis in BALB/c Mice for Vaccine Development"

_pharmaceutics, 2022, doi:10.3390/pharmaceutics14101989_

Round 1
Reviewer 1 Report (Previous Reviewer 2)
The author revised the paper according to the reviewer's suggestion and corrected the errors. Unfortunately, due to resource constraints, the author could not continue to carry out further research to find valuable pure antigenic proteins from total secretions. Overall, this study provides a certain reference for further understanding of the immunogenic secretory proteins and immune pathogenic mechanism of Candida glabrata, and has certain scientific significance and value.
Author Response
Response to the reviewer comments
Dear Editor
MDPI (Pharmaceutics)
Thank you for giving us the opportunity to submit a revised draft of our manuscript entitled, Characterization of the secretome of pathogenic Candida glabrata and their effectiveness against systemic candidiasis in BALB/c mice for vaccine development, to Pharmaceutics. We appreciate the time and effort that you and the reviewers have dedicated to providing your valuable feedback on our manuscript. We are grateful to the reviewers for their insightful comments on our paper. We have been able to incorporate changes to reflect all the suggestions provided by the reviewers. The changes within the manuscript are highlighted.
Here is a point-by-point response to the reviewers’ comments and concerns.
#1 Comments and Suggestions for Authors
The author revised the paper according to the reviewer's suggestion and corrected the errors. Unfortunately, due to resource constraints, the author could not continue to carry out further research to find valuable pure antigenic proteins from total secretions. Overall, this study provides a certain reference for further understanding of the immunogenic secretory proteins and immune pathogenic mechanism of Candida glabrata and has certain scientific significance and value.
Response: Thank you sincerely for reviewing our manuscript and understanding the challenges. We will in future research consider the reviewer’s comment. Reviewer’s contributions and appraisals are highly appreciated.
Reviewer 2 Report (New Reviewer)
This is an interesting research study, which dealt with the characterization of C. glabrata secretome and their effectiveness against systemic candidiasis for vaccine development. While the work is a significant contribution to the literature, I do have some concerns which should be considered before the acceptance of the manuscript.
1. Antifungal susceptibility profiling: why did the authors measure MIC for only 2 antifungal agents? I think they should check the MIC for mere antifungals. Furthermore, according to CLSI M27, the MIC should be determined after 24h not after 48h, as described by the authors. I think they should report their results after 24 h or try the experiment if the 24 h results are not in hand. CLSI M27 did not describe the MFC protocol, therefore the authors should add a reference for the MFC experiment.
2. L198-L199 you mentioned “three commonly used antifungal drugs representing the three main antifungal classes” this is not correct, as you only tested two antifungals. Please correct and as I mentioned please try to increase the number of tested antifungals.
3. I think the methods from 2.4 to 2.8 should be collected under a single method (for example, protein extraction and LC-MS analysis) and structured as you performed in 2.3 and 2.9.
4. Some methodology required to include reference(s) such as biofilm formation, extraction of secreted proteins from growth media, and LC-MS data acquisition and searching.
5. The results of table 1 should be modified according to the MIC after 24 h as I explained previously.
6. Please include the full name of the abbreviations under table one.
7. A full description of Fig 1 is essential in its legend. Please also confirm your finding by adding arrows or arrowheads in Fig. 1.
8. L 208: C. glabrata should be italic, and please confirm throughout the manuscript and the references.
9. In fig 2 you included only the results of proteinase activity, please include also the results of phospholipase activity even negative results.
10. In Table 2 legend, please include the difference between bold and non-bold written proteins. Also in the discussion (L275), you mentioned the meaning of boldly written proteins, but without highlighting non-bold written proteins, please clarify.
11. L264 to L266: what is the significance of this sentience?
12. Fig 4 is very confusing, what do you mean by Day 1, Day 2, and Day 21 on the right side of the figure? I think figure 4 should be moved to materials and methods (In vivo studies)
13. Why do animals with signs of distress after 24h and 48h were euthanized? Why you did not wait until their death? Why do you exclude the self-recovery of the animals? Is this common in such kinds of studies? If so, please support with reference(s) or explain.
14. Several questions were raised from Fig 6:
A- What do you mean by fungal burden was calculated after 24 h and 48 h? According to the materials and methods (L191), the colonies were counted only after 48 h, please explain.
B- In the figure legends (also in Fig 5) you used the abbreviation SPE, but in the figure you used the abbreviation SP, please standardize.
C- You did not mention exactly on what date the animals were euthanized for fungal burden calculation and for histopathology. Please clarify.
15. In Figure 7 legends, please explain what arrows indicate for?.
16. How about the gross pathology of the organs, I think it is important to show the differences between the 3 groups based on the macroscopic manifestations of infection in target organs.
17. L 361 to 369 and Table 3, you are dealing with the comparison between organ weights between 3 groups without statistical analysis. This is not an accurate comparison, you should perform a statistical analysis.
18. In table 3 you mentioned the average weight of the organ, how many replicates? Why you did not include the SE or SD of the average weight?
19. L 372 to L373, please discuss the reasons for the difference between your finding and Hirayama and co-workers (2020) findings.
Round 2
Reviewer 2 Report (New Reviewer)
I think the manuscript after revision is much improved.
1- Please, change method number 2.3.4. to 2.4., 2.3.5. to 2.4.1, 2.3.6. to 2.4.2, 2.3.7. to 2.4.3, 2.3.8. to 2.4.4, 2.3.9. to 2.4.5, 2.3.10. to 2.5., 2.3.11. to 2.6., 2.3.12. to 2.7. and 2.3.13. to 2.8.
2- L372: please change "a significant difference" to "a difference" as Fig 8 doesn't show any statistical differences and no statistical analysis.
Author Response
Please see the attachment

This manuscript is a resubmission of an earlier submission. The following is a list of the peer review reports and author responses from that submission.
Round 1
Reviewer 1 Report
The subject proposed by the authors is interesting. Indeed, candidiasis is becoming more and more important as an infectious pathology. Because of their little importance until now, they are often ignored. They can be slow growing. These infections are often fatal for immunocompromised people.
The authors propose to set up a feasibility study for a vaccination with targeted, immunogenic proteins and importance in the development of pathologies.
After a search for the best candidates by targeting mainly secretome proteins, the authors used the HPLC method to select the most interesting and immunogenic candidates.
In the Materials and Methods section, the authors do not specify the culture conditions (temperature and atmosphere) and the concentration of yeast used for the susceptibility profile.
The authors fix the biofilm with glutaraldehyde which is a good fixative allowing to preserve the structures. However the fixation time of 2H is too high. Please justify this fixation time.
Table 1 the number of trials is not specified and there is no standard deviation.
For the membership the unit and the number of trials is not specified.
Figure 8: The photos of the organs do not provide important information.
The results obtained are interesting but incomplete. The studies more push the immunogenic capacities of the proteins with the demonstration of immune reactions in mice by the demonstration of presence of specific antibodies and/or production of specific immune cells.
Ref 27, 37 the date is not in bold
Reviewer 2 Report
This study examined the presence of immunogenic proteins in the secretome of C. glabrata ATCC2950 using mass spectrometry. Informatics analysis was used to predict the possible protective antigens present in secretions, and the protective power of total secreted proteins against C. glabrata-infected mice was preliminarily determined. The research results provide a certain reference for further understanding of the immunogenic secretory proteins and immunopathogenesis of C. glabrata. Unfortunately, the article only shows the names of 33 antigenic proteins obtained by secretome sequencing and basic informatics analysis, and does not experimentally identify these proteins. In evaluating the protective ability of secreted proteins against hematogenous disseminated infection caused by C. glabrata, crude extracts of secreted proteins with complex components were used, why not find the most valuable antigenic proteins from these mixtures, and further Study its protective abilities?
Comments and Suggestions:
1. In the abstract section, the author believes that C. glabrata is less sensitive to azoles(Line20), the cited references also show that C. glabrata is inherently resistant to azole antifungals(Line50). But the C. glabrata strains used in this article are very sensitive to fluconazole, especially the clinical strains are very sensitive, the MIC value is only 0.062-0.125µg/ml. The effect of fluconazole is even more significant than that of amphotericin B(Line200), how to explain this phenomenon?
2. Determination of the secretome of C. glabrata by LC-MS/MS, which has been studied before. Moreover, 21 of the 33 antigenic proteins predicted in this study have been characterized. Although some new possible antigenic proteins have been found, these proteins have not been experimentally verified in the paper. Where is the innovation of this research?
3. How many repetitions of the secretome assay samples? The text needs to explain.
4.Line 177 lethal intravenous injection ,Write clearly that it is through the tail vein.
5.In the biofilm formation experiment, in general, the mature biofilm structure of Candida is relatively dense, composed of a large number of pseudohyphae and extracellular matrix. However, in Figure 1, Candida was basically still in the yeast state, no hyphae were seen, only some cells gather together, and no obvious biofilm structure was seen.
6. Line 346 In Figure 7, the histological section is not marked with magnification.
7. In the experiment of fungal load measurement of mouse organs, the bacterial load of liver, spleen, kidney and lung in the infected group was not very different (Fig. 6), but in the pathological section, the bacterial load of kidney in the infected group was significantly higher than that of liver, spleen and lung (Fig. 7), and the colony count and tissue section results were inconsistent. How to explain this phenomenon?
8. References are too old, only 17% in the past 3 years; some reference formats are not standardized and need to be further revised, such as Line495, 541, 555.